# Propose, Solve, Verify: Self-Play Through Formal Verification

Alex Wilf [1]  Pranjal Aggarwal [1]  Bryan Parno [1]
Daniel Fried [1]  Louis-Philippe Morency [1]  Paul Pu Liang [2]  Sean Welleck [1]

## Abstract

Training models through self-play alone (without any human data) has been a longstanding goal in AI, but its effectiveness for training large language models remains unclear, particularly in code generation, where rewards based on unit tests are brittle and prone to error propagation. We study self-play in the verified code generation setting, where formal verification provides reliable correctness signals. We introduce PROPOSE, SOLVE, VERIFY (PSV), a simple self-play framework where formal verification signals are used to create a proposer capable of generating challenging synthetic problems and a solver trained via expert iteration. We use PSV to train PSV-VERUS, which across three benchmarks improves pass@1 by up to 9.6× over inference-only and expert-iteration baselines. We show that performance scales with the number of generated questions and training iterations, and through ablations identify formal verification and difficulty-aware proposal as essential ingredients for successful self-play.

## 1. Introduction

A longstanding goal of AI research is to enable *self-play*, in which a system learns by creating its own curriculum without human supervision (Silver et al., 2017; Sukhbaatar et al., 2018). In the context of large language models (LLMs), an emerging family of *proposer-solver* algorithms holds the promise of improving LLMs through self-play (Haluptzok et al., 2023; Zhao et al., 2025). In this setup, a *proposer* model generates problems for a *solver* model to train on, and the solver's capabilities inform the proposer's curriculum. If one can reliably verify the correctness of the solver's outputs, one can train the solver with reinforcement learning and design curricula that adapt to the solver's capabilities.

[1]Carnegie Mellon University, School of Computer Science [2]MIT. Correspondence to: Alex Wilf <awilf@cs.cmu.edu>.

*Proceedings of the 43rd International Conference on Machine Learning*, Seoul, South Korea. PMLR 306, 2026. Copyright 2026 by the author(s).

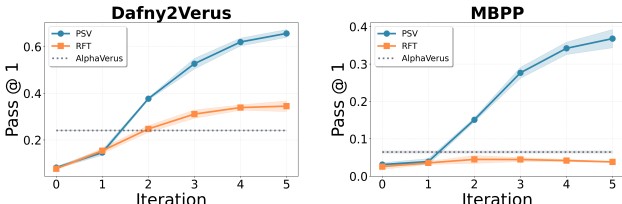

*Figure 1.* *Propose, Solve, Verify (PSV)* uses a formal verifier to enable a self-play loop. We use PSV to train PSV-VERUS, which starts from the same translated seed corpus as AlphaVerus (Aggarwal et al., 2024) but expands it by proposing new problem specifications and training on verified solutions. PSV-VERUS improves across self-play iterations, opening a substantial gap over AlphaVerus or training on the seed corpus alone (RFT).

Thus a fundamental challenge in realizing proposer-solver self-play is *verification*. An imperfect verifier can be exploited by the solver, and hence corrupt the solver's training and any curricula that depend on the solver's performance. Indeed, successful early attempts have come in easy-to-verify domains such as formal theorem proving (Dong & Ma, 2025) or easy-to-check synthetic tasks (Haluptzok et al., 2023; Dong et al., 2025; Zhao et al., 2025), leaving self-play's scope of impact for LLMs unclear.

We study self-play for code generation, a domain where verification typically relies on unit tests (Liu et al., 2023). A fundamental limitation is that unit tests only cover a limited set of cases, so programs can pass them and still be incorrect (Xin & Reiss, 2017; Liu et al., 2023; Yuan et al., 2024). This can lead the solver to optimize for passing tests rather than solving the underlying task, and errors can propagate across iterations (Lin et al., 2025; Baker et al., 2025). Hence self-play in code generation remains an open problem.

We introduce PROPOSE, SOLVE, VERIFY (PSV), which uses *formal verification* to unlock a new paradigm for self-play in code generation. The core idea is for a proposer to generate problems in the form of *formal specifications*—statements that mathematically describe all possible permitted inputs plus the desired output behavior of a program with respect to those inputs—and for a solver to try to generate programs that meet the specifications. The key benefit is that formal verification provides a *sound* reward signal: if the verifier accepts the program, then the program is guar-

## Python: Traditional Unit Tests

Signature
```
def sort(arr):
```

Implementation
```
    for i in range(len(arr)):
        ...
```

Unit Tests
```
assert sort([1,2,3])==[1,2,3]
assert sort([])==[]
assert sort([3,2,2,1])==[1,2,2,3]
...
```

## Verus: Formally Verified Code

Spec
```
fn sort(s: &Vec<i32>) -> (ret: Vec<i32>)
    ensures
        forall|i: int, j: int| 0 <= i < j < ret@.len() ==>
        ret@[i] <= ret@[j],
        ...
{
```

Implementation & proof
```
let mut sorted = s.clone();
...
proof { assert(old_sorted.to_multiset() =~= sorted@.to_multiset());}
...
```

*Figure 2.* **Test cases vs. formal verification**. On the right, given any input that satisfies the specification's preconditions (here, just the type constraint of $s$), a program that meets the specification according to the formal verifier is guaranteed to produce an output that satisfies the postcondition (here, the ensures clause saying that the vector is sorted). In self-play with verified code generation, a proposer produces the function spec (including pre- and postconditions), and the solver produces the program and proofs given a specification. This guarantee stands in contrast to supervision based on LLM-generated unit tests (shown on the left), which can be incomplete (Yang et al., 2025b) and reward hackable (Baker et al., 2025).

anteed to satisfy its specification for all inputs. Thus unlike unit tests, formal verification prevents incorrect solutions from entering the self-play loop.

We apply PSV to Verus (Lattuada et al., 2023), a framework that enables formal verification of Rust programs and has attracted recent interest for LLM code generation (e.g., Yang et al. (2025a); Aggarwal et al. (2024)). We design a *difficulty-aware* proposer that adapts the difficulty of proposed specifications based on the solver's current pass rates. We initialize self-play from the same translated corpus used by AlphaVerus (Aggarwal et al., 2024), which originates from human-written problems. PSV then expands this corpus by alternating between proposing new specifications, training the solver on verified solutions, and using the solver's performance to guide the next round of problem proposal. As shown in Figure 1, the PSV self-play loop yields substantial gains: our model, PSV-VERUS, improves across iterations and substantially outperforms both training on the seed corpus alone (RFT) and AlphaVerus, which relies on the same seed corpus but without self-play.

We show that PSV-VERUS's performance scales with the number of iterations and generated questions, and empirically identify that verification of solutions and the difficulty-awareness and diversity of the proposal are key factors for self-play. Coupled with recent successes in formal theorem proving (Dong & Ma, 2025), our work points to formal verification as a promising frontier for LLM self-play. In summary, our contributions are as follows:

1. We introduce PROPOSE, SOLVE, VERIFY (PSV), a self-play algorithm for code generation that leverages formal verification, and propose a difficulty-aware proposer based on in-context learning.

2. We train PSV-VERUS, which outperforms the previous state-of-the art AlphaVerus (Aggarwal et al., 2024) and

Expert Iteration (Singh et al., 2024) on verified Rust programming.

3. We show that performance scales with question budget and iterations, and identify proposal coverage and difficulty as factors driving self-play.

4. We release our code and models.[1]

## 2. Related Work

**Verified code generation.** Verified code generation asks solver models to generate both programs and machine-checkable proofs that verifiers can deterministically and mechanically check against a specification. For example, in Figure 2, the implementation and proof must satisfy the postconditions described in the ensures block of the spec over *all possible inputs and outputs to the function*. Unlike unit-tests, which are hard for LLMs to generate and vulnerable to reward hacking (Yang et al., 2025a; Baker et al., 2025), formal verification provides a binary guarantee of correctness. This makes the setting attractive for safety-critical domains (Klein et al., 2014) and for self-improving AI that requires verification of novel questions. Recently developed formal verification languages such as Verus (Lattuada et al., 2023) bring SMT-backed proofs into mainstream systems languages such as Rust, making formal verification possible in real world, modern programming domains. A key obstacle to model performance on these tasks is data scarcity (Aggarwal et al., 2024), which makes synthetic corpus generation under formal verifier feedback especially promising. Most related to our setting, recent work uses a verifier to synthesize verified-code training data: AutoVerus (Yang et al., 2025a) and SAFE (Chen et al., 2026) generate proofs for a *fixed* set of existing Rust programs without an adaptive proposer. Unlike these, PSV-VERUS

---
[1] https://github.com/abwilf/psv

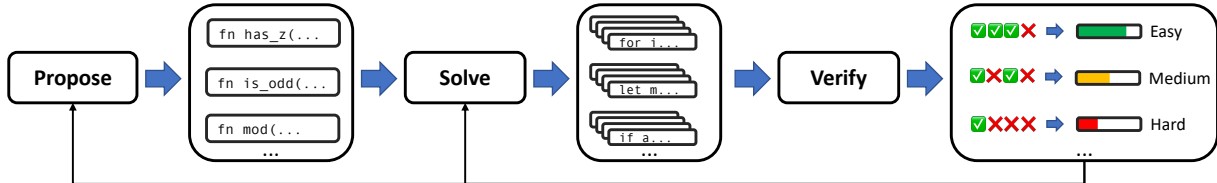

*Figure 3.* PROPOSE, SOLVE, VERIFY (PSV) is an iterative self-play approach for formally verified code generation. At each iteration $t$, the solver $S_{\theta_t}$ attempts to solve the current set of specifications $x_{i,t}$, producing candidate solutions that are checked using formal verification, which *mathematically guarantees* their correctness. Verified solutions are used to update the solver via rejection fine-tuning (RFT), while a proposer $P_{\phi,t}$ generates new, challenging specifications using the pass rate (how many candidate solutions passed verification for each problem) as a proxy for what kinds of problems the solver $S_{\theta_t}$ finds difficult.

*proposes new specifications* via a difficulty- and diversity-aware proposer that co-evolves with the solver, and trains end-to-end for both the implementation and its proof.

**Self-improving AI systems.**   Self-play game playing algorithms such as AlphaZero have shown that AI systems can surpass human performance without human labels (Silver et al., 2017). Yet porting this paradigm to language, in particular reasoning, has been challenging: the action space is vast, and verifiers are often weak instead of deterministic (e.g., the rules of the game define a win/loss). Recent work has made progress on self-improving language models with rejection-finetuning (Singh et al., 2024; Zelikman et al., 2022) and reinforcement learning from verifiable rewards (RLVR) (Shao et al., 2024; DeepSeek-AI et al., 2025; Wen et al., 2025) on fixed human-generated datasets. Yet human-generated reasoning datasets are limited and difficult to scale (Liu et al., 2025), motivating the creation of synthetic data for training reasoning capabilities.

**Self-play reasoning systems.**   Self-play systems leverage an intuition that there is a "gap" between what AI can solve and verify (Song et al., 2024), meaning that systems could self-improve by proposing novel problems and training a "solver" model on verified solutions. This has been applied to reasoning games (Cheng et al., 2025b), code (Rozière et al., 2023; Wei et al., 2024; Haluptzok et al., 2023), and math (Shah et al., 2024). A recent line of work has also proposed *recursively* self-improving, by training the question *proposer* as well to output difficult problems for the current solver (Li et al., 2024; Huang et al., 2025). This has been applied to formal math (Dong & Ma, 2025), coding (Zhao et al., 2025; Lin et al., 2025), tool use (Zhou et al., 2025), alignment (Cheng et al., 2025a; Zhu et al., 2025), and general language tasks (Kuba et al., 2025). As strong verifiers are important to self-play (yet often limited in applicability) (Saad-Falcon et al., 2025), we investigate the verified coding domain in Verus (Lattuada et al., 2023) where SMT-backed verification provides a sound guarantee over a Turing-complete language (Rust).

## 3. PROPOSE, SOLVE, VERIFY (PSV)

PROPOSE, SOLVE, VERIFY (PSV) leverages a formal verifier to enable a self-play loop. The core idea is to iterate between proposing specifications, attempting to solve the specifications, and using the formal verifier's feedback to improve the solver and inform the next round of problem proposal. Over time the solver trains with an evolving pool of challenging and diverse problems, yielding improvements in the solver's capabilities. We describe each component in detail below.

**Problem setting.**   We consider the problem of verified code generation. In this problem, the input is a *formal specification* $x \in \mathcal{X}$, and the output is code $y \in \mathcal{Y}$. A verifier $v(x,y) \to \{0,1\}$ checks whether the code meets the formal specification. If so, the verifier returns 1 and otherwise it returns 0. The specification and code are written in a language that supports verification, such as Verus (Lattuada et al., 2023) which supports verification for a subset of Rust code, or Dafny (Leino, 2010). The code $y$ contains an implementation and any additional proof code (e.g., "loop invariants") that is necessary for the program to pass the verifier. Given a specification $x$, the goal is to produce code $y$ that passes the verifier, $v(x,y) = 1$. The key property that we leverage is that the Verus verifier is sound with respect to specifications; i.e., if $v(x,y) = 1$ then the program $y$ meets the specification $x$. See Appendix F for additional background.

While we consider formally verified code generation here, in principle our methods apply to any problem with a sound verifier; i.e., if the verifier $v(x,y)$ returns 1, then $y$ is a correct output for problem $x$. Applying PSV to other such problems is left for future work.

### 3.1. PROPOSE, SOLVE, VERIFY (PSV)

PSV is an algorithm that runs for iterations $t \in \{0 \dots T-1\}$ and consists of a proposer model $P_{\phi_t}$ and a solver model $S_{\theta_t}$, working together to generate and solve new problems, supervised with help from the verifier.

At the first iteration ($t = 0$), PSV receives a set of seed problem specifications $X_t = \{x_1, x_2, \ldots, x_{N_0}\}$, and the solver attempts to solve them. That is, given a specification $x_i$, the solver produces candidate outputs $y_{i,t}^1, \ldots, y_{i,t}^{k_{trn}}$, where $k_{trn}$ is a hyperparameter governing the number of solver attempts per specification. Each output has an associated verification outcome $v_{i,t}^j \in \{0, 1\}$ obtained by running the verifier on $x_i, y_{i,t}^j$. Over all specifications $x \in X_t$, this yields a data pool $D_t = (X_t, Y_t, V_t)$.

The solver is then trained using the data pool, yielding a new solver $S_{\theta_{t+1}}$. Finally, the proposer $P_{\phi_t}$ is updated using the data pool and is used to generate a new set of $B$ specifications that are added to the existing set, so that $X_{t+1} = X_t \cup \{x_{N+1}, x_{N+2}, \ldots, x_{N+B}\}$. The process iterates for a fixed number of iterations $T$.

Next, we describe the specific design choices for the solver, learning algorithm, and proposer that we use in our experiments, including our *difficulty-aware* proposer. We summarize PSV in Algorithm 1.

---

**Algorithm 1** PROPOSE, SOLVE, VERIFY (PSV)

---

**Require:** Seed specifications $X_0$, solver $S_{\theta_0}$, proposer $P_{\phi_0}$, question budget $B$, verifier $V$, number of iterations $T$
    **for** $t = 0$ **to** $T - 1$ **do**
        $\{Y_t, V_t\} \leftarrow$ SOLVE$(S_{\theta_t}, X_t, V)$
        $D_t \leftarrow (X_t, Y_t, V_t)$
        $S_{\theta_{t+1}} \leftarrow$ TRAIN$(S_{\theta_0}, D_t)$
        $P_{\phi_{t+1}} \leftarrow$ UPDATE$(P_{\phi_0}, D_t)$
        $X_{t+1} \leftarrow X_t \cup$ PROPOSE$(P_{\phi_{t+1}}, D_t)$
    **end for**

---

**Solving.** At each iteration $t$, PSV produces candidate solutions for each $x_i \in X_t$ by sampling from the solver model $S_{\theta_t}$ with temperature 0.8, yielding $k_{trn}$ candidate solutions for each: $y_{i,t}^1, \ldots, y_{i,t}^{k_{trn}}$, along with binary verification outcomes $v_{i,t}^1, \ldots, v_{i,t}^{k_{trn}}$. We follow AlphaVerus's few shot prompting setup (Aggarwal et al., 2024) with a 1-shot example. We found that while increasing the number of few-shot examples improved performance, it also increased runtime substantially because solving is the most computationally intensive part of the pipeline, and the method was improving without more few-shot examples.

**Rejection-Finetuning.** The solver is updated through rejection fine-tuning (RFT) (Singh et al., 2024), where the base model is fine-tuned only on verified solutions. Formally, we construct the training data for a single iteration of RFT as

$$D_t^* \leftarrow \left\{ (x_i, y_{i,t}^j) : i \in D_t, v_{i,t}^j = 1 \right\}$$

We then trim $D_t^*$ to keep a maximum of one solution to each

problem $i$ if there are multiple values of $j$ s.t. $v_{i,t}^j = 1$. The solver is then updated by minimizing the cross-entropy loss:

$$\mathcal{L}(\theta) = -\frac{1}{|D_t^*|} \sum_{(x_i, y_i^j) \in D_t^*} \sum_{k=1}^{|y_i^j|} \log p_\theta(y_i^j[k] \mid x_i, y_i^j[< k])$$

where $\theta$ is initialized from $\theta_0$, $y_i^j[k]$ is the $k$-th token of solution $y_i^j$, and $y_i^j[< k]$ denotes all preceding tokens. Iteratively training on verified solutions in this way can be seen as an offline RL algorithm based on expectation-maximization (Singh et al., 2024). Further details about SFT training can be found in Appendix E.

In principle, different reinforcement learning algorithms could have been used, such as GRPO (Shao et al., 2024). We use RFT because the Verus verifier is guaranteed to be sound but not complete, meaning that advantage-weighted algorithms could incorrectly punish models for correct solutions that were judged to be incorrect. Future work on on-policy RL algorithms for formally verified self-play may investigate more performant RL algorithms in this setting.

**Proposing.** At iteration $t$, proposer $P_{\phi,t}$ generates $B$ new problem specs at target difficulty levels by conditioning on problems and associated difficulty labels. We estimate the difficulty of question $x_i$ at iteration $t$ by the current solver's pass rate on that problem:

$$r_{i,t} = \frac{1}{k_{trn}} \sum_{j=1}^{k_{trn}} v_{i,t}^j$$

We categorize problems into one of four classes based on $r_{i,t}$: EASY, MEDIUM, HARD, and IMPOSSIBLE:

$$\text{difficulty}(x_i, t) = \begin{cases} \text{EASY} & \text{if } r_{i,t} \geq \tau_E \\ \text{MEDIUM} & \text{if } \tau_M \leq r_{i,t} < \tau_E \\ \text{HARD} & \text{if } 0 < r_{i,t} < \tau_M \\ \text{IMPOSSIBLE} & \text{if } r_{i,t} = 0, \end{cases}$$

where the thresholds $\tau_E$ and $\tau_M$ are hyperparameters for our method.

We then sample $k_{prop}$ problems from $D_t$ to use as examples and ask the proposer to output a question at a target difficulty level. A simplified version of our prompt is below, and the complete prompt is in Appendix A. This mechanism continually refreshes the prompt distribution, updating the proposer to $P_{\phi,t+1}$ through in-context learning and expanding the problem frontier in a difficulty-controlled manner.

*Table 1.* Test-Time Training and Transfer Learning Results. Values show mean pass@k rates with standard deviations as subscripts across 5 random seeds. Best results (statistically significant via paired t-test, $p < 0.05$) are **bolded**.

| Method | Dafny2Verus | | | MBPP | | | HumanEval | | |
|---|---|---|---|---|---|---|---|---|---|
| | pass@1 | pass@5 | pass@10 | pass@1 | pass@5 | pass@10 | pass@1 | pass@5 | pass@10 |
| **Inference Only** | | | | | | | | | |
| Base Model | $8.05_{0.19}$ | $20.68_{0.58}$ | $28.15_{1.03}$ | $0.87_{0.12}$ | $4.02_{0.45}$ | $7.53_{0.77}$ | $4.55_{0.23}$ | $10.28_{0.42}$ | $12.86_{0.37}$ |
| AlphaVerus | $24.06_{0.16}$ | $52.42_{0.21}$ | $63.44_{0.29}$ | $6.48_{0.29}$ | $18.36_{0.72}$ | $24.57_{1.02}$ | $7.24_{0.10}$ | $15.38_{0.36}$ | $18.02_{0.54}$ |
| **Transfer Learning** | | | | | | | | | |
| RFT | — | — | — | $10.99_{1.10}$ | $26.03_{1.05}$ | $31.19_{0.79}$ | $10.99_{0.45}$ | $18.07_{0.50}$ | $20.02_{0.54}$ |
| PSV-VERUS | — | — | — | $\mathbf{25.25}_{2.24}$ | $\mathbf{38.32}_{1.80}$ | $\mathbf{41.51}_{1.60}$ | $\mathbf{16.18}_{1.30}$ | $\mathbf{21.61}_{1.50}$ | $\mathbf{23.13}_{2.06}$ |
| **Test-Time Training** | | | | | | | | | |
| RFT | $34.46_{2.29}$ | $57.12_{1.86}$ | $63.40_{1.06}$ | $3.83_{0.19}$ | $14.56_{0.60}$ | $22.64_{1.04}$ | $5.56_{0.41}$ | $12.46_{0.35}$ | $14.80_{0.34}$ |
| PSV-VERUS | $\mathbf{65.63}_{1.70}$ | $\mathbf{78.04}_{2.31}$ | $\mathbf{80.06}_{2.35}$ | $\mathbf{36.78}_{2.45}$ | $\mathbf{51.22}_{1.65}$ | $\mathbf{53.67}_{1.49}$ | $\mathbf{19.07}_{1.02}$ | $\mathbf{23.42}_{1.20}$ | $\mathbf{25.16}_{1.47}$ |

```
Proposer In-Context Learning

I would like you to output a problem that is
of {difficulty} level. Here are some examples
of problems and how difficult they were for
the model.

Problem 1: EASY
{problem spec}
Problem 2: HARD
{problem spec}
...

Now it's your turn! Please describe your
reasoning, then output a new problem that is
{difficulty}.
```

After inference, specifications are parsed, deduplicated, and filtered through a spec verifier which ensures that the spec defines a valid set of mathematical pre- and post-conditions. This uses a form of Verus that ignores the implementation body – details are in Appendix D.

## 4. Experiments

We study self-play verified code generation in the Verus language (Lattuada et al., 2023).

**Datasets and evaluation.** We evaluate our approach using three Verus datasets. **Dafny2Verus** (271 problems) is a translation-derived corpus adapted from the AlphaVerus (Aggarwal et al., 2024) pipeline, providing a diverse set of formally verified Dafny problems translated into Verus. **MBPP**-Verified (78 problems) is based on the MBPP dataset (Austin et al., 2021), following the verified adaptation process described in (Yang et al., 2025a; Misu et al., 2024), where each task includes executable Rust+Verus specifications and proofs. **HumanEval**-Verified (85 functions across 49 original programs) is drawn from an open-source HumanEval-Verus effort (Contributors, 2024), which translates Python HumanEval tasks (Chen et al., 2021) into Verus. Each multi-function program is decomposed into

independently verifiable functions, ensuring all necessary dependencies are preserved. All datasets require generating Rust+Verus code that passes the Verus verifier (Lattuada et al., 2023). We will omit the *-Verified from the names for brevity throughout the rest of the paper. We adopt the widely used Pass@$k$ metric (Chen et al., 2021): for $n$ samples and $c$ verified successes, pass@$k = 1 - \binom{n-c}{k}/\binom{n}{k}$. We measure Pass@1, 5, and 10, and run our final evaluation with $n = 100$ to provide a strong, unbiased estimate of the evaluation metrics.

**Settings.** We study our method's performance in two settings. In a setting we refer to as *transfer learning*; we set $X_0$ to the set of questions in Dafny2Verus and report performance on MBPP and HumanEval as a held out test set. In a setting that we refer to as *test-time training (TTT)* (Sun et al., 2020), we set $X_0$ to the set of questions in each dataset, and run our method on that dataset alone. It is worth noting that human-written solutions are *never trained on*.

**Baselines.** We compare to AlphaVerus (Aggarwal et al., 2024) (no treefinement version), which is the previous SOTA method for verified code generation, based on prompting with 50 in-domain exemplars. We re-implement AlphaVerus on the *same* base model as PSV-VERUS (Qwen2.5-Coder-3B-Instruct) using its publicly available code rather than using the original authors' released model (which is based on a different, much larger base model). This isolates the effect of the training methodology from that of the base model. We also compare to iterative RFT (Yuan et al., 2023; Singh et al., 2024), which performs expert iteration without proposing new problems. We train and evaluate RFT with the same parameters as our method, excluding any parameters related to question proposal.

**Hyperparameters and design choices.** Our hyperparameters and design choices were selected to provide a proof of concept of our idea at a minimal compute scale. In the current setting, the main experiment (Table 1) takes around

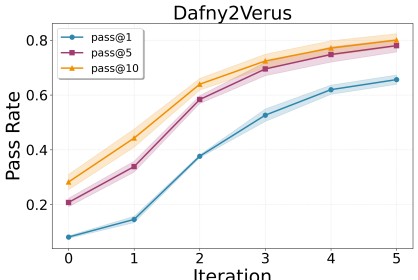 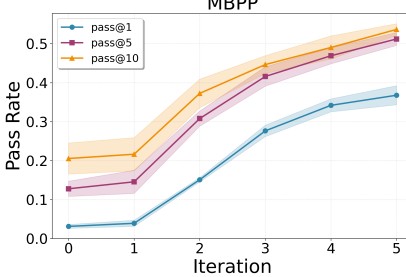 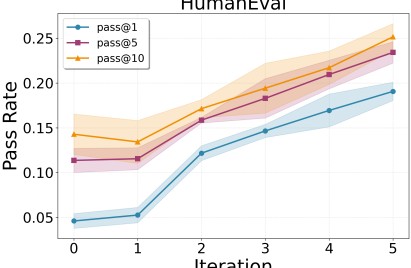

*Figure 4.* PSV-VERUS performance across three test-time training tasks, plotting Pass@1, 5, and 10 across iterations of the algorithm, with clouds representing ± 1 standard deviation across 5 random seeds.

24 hours on a single machine with 8xL40s GPUs. We use the Qwen2.5-Coder-3B-Instruct model as our base because it was the strongest open-weight code model at the 3B size. We use SGLang (Zheng et al., 2024) with dp-size=8 and temperature 0.8 for all inference sampling because SGLang has large efficiency gains for inference time sampling and can parallelize over multiple GPUs on the same machine easily. We set the lower difficulty threshold $\tau_M$ to 0.2, similar to Dong & Ma (2025) (who used 0.25), and set $\tau_E$ to 0.8. We set $k_{trn}$ to 10 to balance having enough breadth in our search to enable performance improvements and not so much that the computational requirements would become unreasonable to reproduce, and train starting from the base model in each iteration as in Singh et al. (2024). We use $k_{prop} = 12$ and use uniform input and output targets, meaning that we populate the proposer prompt with 3 problems for each difficulty class, and ask for $B/4$ problems of each difficulty class as the output. We set our proposer budget $B = 10,000$ (we justify the choice of uniform targets in Appendix B) and vary the number of iterations $T$ and proposer budget $B$ in our analysis (§ 6).

## 5. Results

We investigate PSV-VERUS's performance on our two experimental settings: *transfer learning* and *test-time training*. Table 1 presents our main results on three benchmarks: Dafny2Verus, MBPP-Verified, and HumanEval-Verified. PSV-VERUS consistently outperforms both the AlphaVerus and the RFT baselines across all metrics.

**Transfer learning** PSV-VERUS achieves strong cross-domain generalization, scoring 25.25% pass@1 on MBPP and 16.18% on HumanEval, outperforming AlphaVerus (6.48%, 7.24%) and RFT (10.99%, 10.99%) by 3.90× and 2.30× on MBPP and 2.24× and 1.47× on HumanEval, respectively. These results demonstrate that training on Dafny2Verus generalizes robustly across domains, yielding substantial gains on realistic natural-language programming benchmarks.

**Test-time training** On Dafny2Verus pass@1, PSV-VERUS achieves 65.63%, compared to AlphaVerus's 24.06% (a 2.73× improvement) and RFT's 34.46% (a 1.90× improvement). On MBPP pass@1, PSV-VERUS reaches 36.78%, outperforming AlphaVerus's 6.48% (5.68×) and RFT's 3.83% (9.61×), and on HumanEval pass@1, PSV-VERUS achieves 19.07%, compared to AlphaVerus's 7.24% (2.63×) and RFT's 5.56% (3.43×). On Dafny2Verus pass@10, PSV-VERUS outperforms RFT by 1.26× and AlphaVerus by 1.26×. Averaged across all benchmarks and metrics, PSV-VERUS obtains 48.12%, compared to 25.43% for RFT and 25.55% for AlphaVerus, representing mean improvements of 1.89× and 1.88×, respectively. Performance graphs for each task are in Figure 4, where the trends of improving over iterations become clear.

## 6. Analyses and Ablations

In the sections that follow, we investigate how our method scales with the number of generated questions (§6.1) and iterations of the algorithm when the number of proposed questions is held constant (§6.2) – experiments that articulate the scaling properties of our method with respect to increasing compute. We also analyze the importance of formal verification and factors impacting the effectiveness of the question proposal through ablation experiments in §6.3 to better understand their impact on self-play performance.

### 6.1. Scaling with Questions Per Iteration

The first scaling factor we consider is the number of questions per iteration $B$, because it can be scaled in parallel across many machines simultaneously without any blocking factors. Results for the test-time training setting are shown in Figure 5.

On Dafny2Verus, scaling from 4k to 32k questions per iteration yields consistent improvements: pass@1 increases from 59.5% to 74.3% (**+25% relative**), pass@5 from 73.8% to 82.6% (**+12%**), and pass@10 from 75.8% to 83.6% (**+10%**). On MBPP, scaling from 1k to 32k questions yields pass@1 improvements from 22.3% to 44.3% (**+98%**), pass@5 from

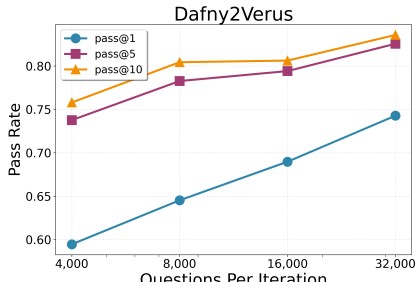 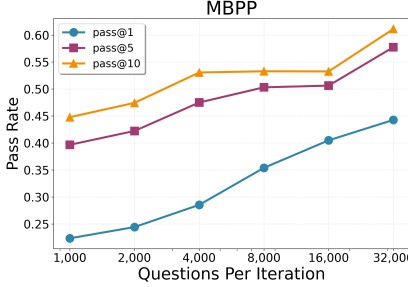 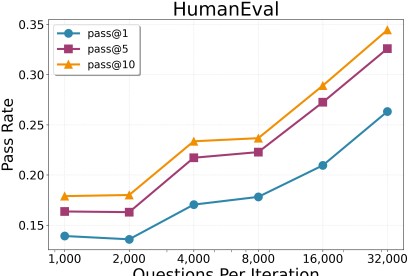

*Figure 5.* Test-time training performance scales with the number of questions per iteration across all three benchmarks.

39.7% to 57.8% (**+46%**), and pass@10 from 44.8% to 61.1% (**+36%**). HumanEval shows similar gains: pass@1 from 13.9% to 26.3% (**+89%**), pass@5 from 16.4% to 32.6% (**+99%**), and pass@10 from 17.9% to 34.5% (**+93%**). We also observe positive scaling behavior for transferring from Dafny2Verus to MBPP, though more modest gains for transferring to HumanEval; see Appendix C.

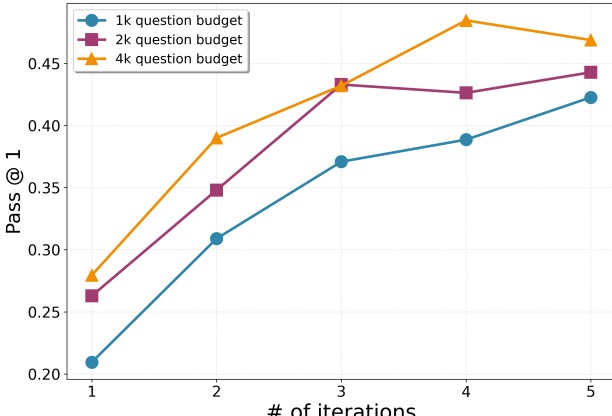

*Figure 6.* Given a fixed total question budget (1, 2, or 4k questions) spread across differing numbers of iterations, we find that running PSV-VERUS with more iterations improves performance. This plot depicts the final Dafny2Verus pass@1 result for different question budgets across number of total iterations used. For example, the 1k line at the first tick mark on the x-axis means that 1000 questions were generated then trained on once, whereas at the last tick mark 200 questions were generated per iteration, separated by updating the solver and proposer.

### 6.2. Iterative Training

Next, we are interested in whether the proposer's budget of generated specifications (i.e., the question budget) should be allocated towards increasing the number of iterations of the method or increasing the number of generated specifications per iteration. We fix the question budget and vary the number of iterations. Figure 6 illustrates a smooth, monotonic improvement as the number of iterations increase. That is, distributing a fixed question budget across multiple of iterations proposing, solving, and updating yields

better results than a single large iteration. For example, in the test-time-training setting on Dafny2Verus at a 1k budget, increasing iterations from 1 to 5 boosts pass@1 from 21.0% to 42.3% (**+102%**). The pattern holds at higher budgets: at 2k, pass@1 rises from 26.3% to 44.3% (**+68%**); at 4k, from 28.0% to 46.9% (**+68%**). This effect extends beyond pass@1: at the 4k budget level, pass@5 improves from 50.6% to 62.9% (**+24%**), and pass@10 improves from 60.0% to 66.7% (**+11%**). In summary, holding the number of proposed specifications constant, increasing the number of iterations boosts success rates, showing that iterative training drives performance improvements in our method.

### 6.3. Ablations

Our self-play algorithm PSV-VERUS has three key components worth examining through ablation analyses – a **verifier** that determines whether specs and solutions are correct, and a proposer that outputs **diverse** questions by *sampling questions* to put in the proposer prompt from the ever-growing dataset and that outputs questions of target **difficulty** by computing and *including labels* of how difficult questions included in the prompt were for the most recent solver. We decided to ablate each of these components and observe their effect on performance in the transfer learning setting. Our results are in Table 2 and described below in detail.

**Verification**   The Verus verifier is used primarily in our algorithm to check whether solutions are valid for a given spec before training on them. We analyze the impact of this by ablating this step and training on all generated solutions, ignoring the verification signal. Our results are shown in Table 2 under (-Verification). We find that removing solution verification hurts performance significantly, leading to relative drops in performance of **51.5% pass@1** and **29.3% pass@5** for Dafny2Verus, **55.2%** and **35.7%** for MBPP, and **52.1%** and **33.5%** for HumanEval.

In addition to performance gains, we also find that verification can be useful for efficiency. We also use verification to filter out candidate specs to make sure they are syntactically compilable before attempting inference on them. This impacts efficiency only, not performance, because the syn-

*Table 2.* Ablation Study. We find that verification of the solution, difficulty-aware proposal prompting, and diversity through proposal context sampling are important to self-play performance. These results report mean pass@k with standard deviation subscripts across 5 seeds.

| | **Dafny2Verus** | | | **MBPP** | | | **HumanEval** | | | |
|---|---|---|---|---|---|---|---|---|---|---|
| | pass@1 | pass@5 | pass@10 | pass@1 | pass@5 | pass@10 | pass@1 | pass@5 | pass@10 | **Avg** |
| PSV-VERUS | $65.63_{1.70}$ | $78.04_{2.31}$ | $80.06_{2.35}$ | $25.25_{2.24}$ | $38.32_{1.80}$ | $41.51_{1.60}$ | $16.18_{1.30}$ | $21.61_{1.50}$ | $23.13_{2.06}$ | 43.31 |
| -Verification | $31.82_{1.42}$ | $55.21_{1.26}$ | $60.69_{1.52}$ | $11.31_{1.46}$ | $24.66_{1.86}$ | $28.86_{2.44}$ | $7.75_{0.71}$ | $14.38_{0.72}$ | $16.39_{0.92}$ | 27.90 |
| -Difficulty | $60.16_{2.67}$ | $74.98_{3.66}$ | $77.73_{3.45}$ | $23.03_{1.51}$ | $36.25_{1.56}$ | $39.73_{1.55}$ | $13.83_{1.12}$ | $20.17_{1.08}$ | $21.92_{1.14}$ | 40.87 |
| -Diversity | $54.95_{1.43}$ | $71.46_{1.15}$ | $74.19_{0.86}$ | $20.28_{1.04}$ | $34.13_{2.16}$ | $37.66_{2.04}$ | $14.25_{0.82}$ | $20.04_{0.74}$ | $21.40_{0.93}$ | 38.71 |

tactically invalid specs are 100% not solvable, so no solution can impact training. We ablated this step and found that the efficiency hit from removing spec verification is substantial: only 47.7% of unique specs were valid on average, so removing spec verification meant that we had to perform pass@10 inference on the remaining 52.3% of specs that were not solvable, a **2.1x increase in inference compute** with no resulting performance improvement.

**Difficulty-awareness** We experiment with removing the difficulty awareness of our proposer by removing the labels for the sampled questions in the proposer prompt. The prompt form is detailed in Appendix G. We found that this -Difficulty ablation causes an average drop of **5.6%** across all metrics. Table 2 shows per-dataset comparisons: the drop is statistically significant for 6 of 9 dataset/metric combinations (paired t-test, $p < 0.05$). Notably, all three Dafny2Verus metrics show significant drops, while some HumanEval and MBPP comparisons do not reach significance (MBPP pass@1, HumanEval pass@5,10).

To understand why, we analyze whether the proposer successfully generates questions at the target difficulty. For PSV-VERUS, easy-targeted questions have mean pass rate 0.55 (std 0.47), medium-targeted questions have mean 0.45 (std 0.42), and hard-targeted questions have mean 0.36 (std 0.39). The different means but substantial overlap between distributions indicates that difficulty prompting provides **partial but incomplete control** over generated question difficulty. It seems that difficulty-aware prompting provides a modest benefit despite its imperfect control over generated question difficulty – an area that future work may find fruitful to improve upon.

**Diversity** We experiment with limiting the diversity of the generated questions by ablating the constantly refreshing proposer prompt with a fixed proposer prompt. We construct this by sampling only once from the seed dataset to get our in-context examples, similar to the fixed prompt in self-instruct pipelines such as Rozière et al. (2023). This ablation causes an average drop of **10.6%** across all metrics. Table 2 show the full results: all Dafny2Verus and MBPP metrics show significant drops (pass@1 drops of 16.3% and 19.7% respectively), while HumanEval pass@5 and pass@10 are

lower but do not reach significance.

The mechanism behind this improvement is increased sample diversity: PSV-VERUS achieves a higher uniqueness rate on average (45.3% vs 29.3%), which leads to 2.48× more unique questions, and 2.26× more solvable questions used for training the solver, enabling the model to learn from a broader distribution of verified solutions.

## 7. Conclusion

This work set out to investigate a central obstacle to proposer-solver self-play algorithms for language models: obtaining a reliable reward signal that prevents error propagation and reward hacking. While prior self-play successes have largely been confined to domains with trivial or limited-applicability verifiers, we showed that *formal verification* in a Turing-complete coding language enables self-play in a realistic code-generation setting by providing a sound, non-exploitable training signal. By introducing PSV and applying it successfully to verified Rust programming, we demonstrated that a model can autonomously expand its training curriculum beyond human-written data and steadily improve its capabilities without human supervision. We show that our algorithm displays smooth scaling behavior with respect to the number of generated questions (§6.1) and iterations (§6.2), making it an exciting first step in understanding the limits of scaling for self-play reasoning training. In analyzing key components of our algorithm's design, we observe the importance of formal verification and difficulty-aware and diverse question proposal (§6.3) in making self-play performant.

## Impact Statement

We believe this work could help usher in a future of self-improving models through self-play. This comes with ethical concerns, but so too does it have the potential to enable a more equitable future where reasoning can emerge in many domains—more dependent on the properties of the generation-verification gap in that domain than on the capability to collect costly human reasoning traces (a capability usually only the highly resourced possess).

## Acknowledgments

This work was also supported in part by the National Science Foundation under Grant Nos. DMS-2434614 and DMS2502281, the Future Enterprise Security initiative at Carnegie Mellon CyLab (FutureEnterprise@CyLab), and AFRL and DARPA under Agreement FA8750-24-9-1000, and by gifts from Amazon, Convergent Research, Microsoft, VMware, and the Beneficial AI Foundation.

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

# A. Full Proposer Prompt

```
Proposer In-Context Learning

I would like you to output a function spec in Verus (Rust) that is of difficulty {difficulty}. A spec
defines a function's inputs, outputs, and returns, and may define preconditions (requires) and
postconditions (ensures) as well as any helpers necessary for defining these attributes.

First you should reason about your answer, then you should output the problem descriptions for each
problem within ```rust ``` tags, so that it can be parsed.

Your solution will take the form:

# Reasoning

< your observations about what makes the example problems easy or hard, and ideas about how to propose
a new problem >

```rust

<function spec you propose>

```

Here are some examples of specs and their difficulties.
{examples}

Now it's your turn! Please output a problem that is **{difficulty}** for the model by either making
problems that are not challenging enough harder, or making problems that are too challenging easier,
or both, in some creative combination. Please enclose your function spec in ```rust ``` tags, so that
it can be parsed. DO NOT copy any of the examples, and DO NOT include any function implementations.
You should END your target function with an open curly brace {{.
```

# B. Input & Target Proposal Settings

*Table 3.* Input/Target Proposal Settings. Mean pass@k with standard errors across 5 seeds. A=All, E=Easy, H=Hard, U=Uniform.

| | Dafny2Verus | | | MBPP | | | HumanEval | | | |
| | pass@1 | pass@5 | pass@10 | pass@1 | pass@5 | pass@10 | pass@1 | pass@5 | pass@10 | **Avg** |
|---|---|---|---|---|---|---|---|---|---|---|
| PSV-VERUS | $65.63_{1.70}$ | $78.04_{2.31}$ | $80.06_{2.35}$ | $25.25_{2.24}$ | $38.32_{1.80}$ | $41.51_{1.60}$ | $16.18_{1.30}$ | $21.61_{1.50}$ | $23.13_{2.06}$ | 43.31 |
| A→H | $64.50_{1.31}$ | $77.97_{2.18}$ | $79.78_{2.34}$ | $21.73_{1.40}$ | $35.85_{1.80}$ | $39.43_{1.64}$ | $13.60_{1.17}$ | $20.14_{0.57}$ | $21.94_{0.57}$ | 41.66 |
| E→H | $54.29_{1.70}$ | $70.86_{1.39}$ | $73.59_{1.54}$ | $16.95_{1.58}$ | $32.98_{1.25}$ | $37.28_{1.01}$ | $14.83_{0.81}$ | $20.42_{0.42}$ | $21.75_{0.60}$ | 38.10 |
| A→E | $58.70_{1.35}$ | $73.12_{1.37}$ | $75.35_{1.54}$ | $21.26_{0.81}$ | $34.66_{1.25}$ | $37.70_{1.37}$ | $15.87_{0.96}$ | $21.51_{0.70}$ | $23.00_{0.60}$ | 40.13 |

We investigated different input and target proposal settings – e.g., using all problem types as the input to the proposer, asking for Uniform problem types as output target, or perhaps asking for only one class of problems as output (e.g., Easy, Hard). We tested three alternatives to our method: All → Hard, Easy → Hard (motivated by Luo et al. (2023)), and All → Easy. We found that our model's setting worked the best (All input problem types → Uniform output targets). This may be partially explained by the ablation in §6.3, where we found that removing sampling of the inputs in the proposer hurt performance. Our results suggest that taking in as much of the seed dataset as possible (not limiting input problems to a specific difficulty type) and using that to generate as diverse a sampling of problems as possible is the best approach.

# C. Transfer Learning Scaling Results

We also evaluate how PSV-VERUS scales in the transfer learning setting, where we train on Dafny2Verus and evaluate on MBPP and HumanEval. On MBPP, scaling from 4k to 32k questions per iteration yields consistent improvements: pass@1 increases from 21.8% to 32.0% (**+47%**), pass@5 from 34.4% to 42.6% (**+24%**), and pass@10 from 37.3% to 44.5% (**+19%**). In contrast, HumanEval shows minimal scaling effects: pass@1 remains essentially flat (14.4% to 14.3%, **-1%**), while pass@5 and pass@10 show marginal gains (+4% and +3%, respectively). This is supported by prior work (Aggarwal et al., 2024) which found limited transfer between Dafny2Verus and HumanEval-Verified.

# D. Spec Compilation

In Verus, specifications define the contract a function must satisfy—preconditions (`requires`) and postconditions (`ensures`). To test whether a specification is well-formed without requiring a correct implementation, we use the `#[verifier::external_body]` attribute. This tells Verus to trust the function signature and specification without verifying the implementation body.

**Example: Finding the Maximum Element**

**Specification Only** (with `external_body` stub)

```
1   use vstd::prelude::*;
2
3   verus! {
4
5   #[verifier::external_body]
6   fn max_element(a: &Vec<i32>) -> (max: i32)
7     requires
8         a.len() > 0,
9     ensures
10        forall|i: int| 0 <= i < a.len() ==> a[i] <= max,
11        exists|i: int| 0 <= i < a.len() && a[i] == max,
12  {
13    assume(false);
14    arbitrary()
15  }
16
17  } // verus!
```

The `external_body` stub uses `assume(false); arbitrary()` to satisfy Verus's type checker without a real implementation. This allows us to verify that the specification itself is syntactically valid and logically consistent, independent of any implementation.

**Complete Verified Implementation**

```
1   use vstd::prelude::*;
2
3   verus! {
4
5   fn max_element(a: &Vec<i32>) -> (max: i32)
6       requires
7           a.len() > 0,
8       ensures
9           forall|i: int| 0 <= i < a.len() ==> a[i] <= max,
10          exists|i: int| 0 <= i < a.len() && a[i] == max,
11  {
12      let mut max = a[0];
13      for i in 1..a.len()
14          invariant
15              forall|j: int| 0 <= j < i ==> a[j] <= max,
16              exists|j: int| 0 <= j < i && a[j] == max,
17      {
18          if a[i] > max {
19              max = a[i];
20          }
21      }
22      max
23  }
24
25  } // verus!
```

The verified implementation includes loop invariants that mirror the postconditions, allowing Verus to prove the implementation satisfies the specification. The invariants maintain that:

1. All elements seen so far are $\leq$ max

2. The current max value exists somewhere in the portion of the array already traversed

## E. Supervised Fine-Tuning (SFT) Training Details

We perform all supervised fine-tuning (SFT) using the SFTTrainer implementation from the Hugging Face trl library.[2] The training setup mirrors standard LoRA-based fine-tuning for decoder-only models, with lightweight adaptation for efficiency on a single GPU. All experiments are conducted using accelerate to scale to distributed configurations, though all runs reported in the main results with the Qwen2.5-3B-Coder model use a single NVIDIA L40S GPU (no DDP).

*Table 4.* Model Configuration

| Parameter | Value |
|---|---|
| max_seq_length | 2048 |
| lora_r | 16 |
| use_quantization | False |
| use_gradient_checkpointing | True |
| per_device_train_batch_size | 1 |
| gradient_accumulation_steps | 8 |
| lora_target_modules | q_proj, k_proj, v_proj, o_proj, gate_proj, up_proj, down_proj |

*Table 5.* Training Setup

| Parameter | Value |
|---|---|
| learning_rate | 2e-4 |
| num_train_epochs | 3 |
| gradient_accumulation_steps | 8 |
| eval_strategy | no |
| use_peft | True |
| lora_alpha | 32 |
| bf16 | True |
| save_total_limit | 1 |

Each run takes approximately 20 minutes at the final iteration of the main experiments (with the most sft data). Checkpoints

---

[2]https://huggingface.co/docs/trl/en/sft_trainer

are saved every 5 steps, retaining only the latest one to minimize storage.

**Reproducibility.**    All code for SFT training, including configuration files and launcher scripts, is available in the project repository for full reproducibility, and is run automatically as a subprocess during the main entrypoint script.

# F. Verification Background

Denote a problem as $x \in \mathcal{X}$, and its associated set of acceptable programs as $Y_x \subseteq \mathcal{Y}$. Here $\mathcal{X}$ is a problem domain and $\mathcal{Y}$ is the set of possible programs. Intuitively, the term acceptable corresponds to the notion of a "correct solution".

Let $Z_x^v \subseteq \mathcal{Y}$ be the set of programs for which a binary verifier $v$ returns 1 for problem $x$, i.e.

$$Z_x^v = \{y \in \mathcal{Y} \mid v(x, y) = 1\}. \tag{1}$$

We say that program $y$ is **verified according to** $v$ when $y \in Z_x^v$.

We refer to $Z_x^v$ as an **accepted set** (for verifier $v$, on problem $x$; we occasionally do not state these distinctions for brevity).

By **soundness** we mean that every program verified according to $v$ is an acceptable program:

$$Z_x^v \subseteq Y_x. \tag{2}$$

Namely, we say that *verifier $v$ is sound for problem $x$* if the relationship above holds, and unsound for problem $x$ if the relationship above does not hold. We say that the *verifier is sound* if it is sound for all problems $x \in \mathcal{X}$. We say that the *verifier is unsound* if it is unsound for at least one problem $x \in \mathcal{X}$.

By **completeness** we mean that every acceptable program is verified according to $v$:

$$Y_x \subseteq Z_x^v. \tag{3}$$

We say that *verifier $v$ is complete for problem $x$* if the relationship above holds, and incomplete if the relationship does not hold. We say that the verifier is complete if it is complete for all problems $x \in \mathcal{X}$. We say that the verifier is incomplete if it is incomplete for at least one problem $x \in \mathcal{X}$.

**Test-case verification is unsound.**    A test suite can be seen as a verifier $v_T$ that returns 1 when a program passes all tests in the suite $T$, and 0 otherwise. It induces a subset of programs that pass all tests in $T$, $Z_x^{v_T} = \{y \in \mathcal{Y} \mid y \text{ passes } T\}$.

When we say that test case verification is unsound, we mean that for some problem $x$, there exists a program $y$ that passes the test cases but is not actually correct:

$$\exists x \in \mathcal{X} \text{ s.t. } Z_x^{v_T} \not\subseteq Y_x. \tag{4}$$

It is easy to construct such an example. For instance, consider the problem of writing a program to check whether an input integer is prime, and a single test that checks if the program returns `True` for input 3. Then a program `return n == 3` would pass the tests, but it is clearly not an acceptable program for the problem. In practice, unsoundness of test-case verification is widely observed in the context of LLM code generation and evaluation (e.g., (Liu et al., 2023)).

**Formal verification is sound with respect to the specification.**    In this setting, we take $\mathcal{X}$ to be the set of specifications in a formal verification language such as Verus or Dafny. In our setting, $x$ is a specification (e.g., preconditions and postconditions), $Y_x$ is the set of programs that satisfy $x$, and $v_F$ is the Verus verification procedure. Verus is an SMT-based verifier: it translates a program and specification into verification conditions and discharges them with an SMT solver. When Verus verifies a program, the result is sound with respect to the specification, up to the trusted computing base (Lattuada et al., 2023):

$$y \in Z_x^{v_F} \implies y \in Y_x. \tag{5}$$

That is, if $v_F(x, y) = 1$ then $y$ satisfies the specification $x$.

However, formal verification tools like Verus or Dafny are incomplete. This is fundamental: no (general-purpose) program analysis for Turing-complete languages can be both sound and complete (Rice, 1953). Hence, for a problem $x$, it may be the case that $Z_x^{v_F} \subset Y_x$ (note the strict subset), meaning that there are acceptable programs that the verifier rejects. This fundamental incompleteness explains why proof annotations are needed; we can think of adding such annotations as converting a program $p \in Y_x \setminus Z_x^{v_F}$ into a program $p' \in Z_x^{v_F}$.

## G. Difficulty Unaware Ablation

The prompt is very similar, but the problem difficulties are chosen randomly.

```
Proposer In-Context Learning

I would like you to output a problem that is of {difficulty} level. Here are some examples of problems
and how difficult they were for the model.

Problem 1
{problem spec}
Problem 2
{problem spec}
...
Now it's your turn! Please describe your reasoning, then output a new problem that is {difficulty}.
```

## H. Analysis of Generated Specifications

Because the specifications used to train the solver are generated by the proposer rather than written by humans, a natural set of questions is whether they are *meaningful*, *diverse*, and *non-trivial*. We analyze each property in turn.

**Meaningful.** Every specification used for training passes a multi-stage filtering pipeline that enforces syntactic validity through spec verification (Appendix D). Of the 10,000 raw proposals generated each iteration, roughly 27.5% pass spec verification—the remaining 72.5% fail to compile or type-check against the Verus compiler—and after deduplication only about 5.4% ($\sim$540) survive as *unique* valid specifications. Every surviving specification therefore defines well-formed `requires`/`ensures` clauses over valid Verus types. As further evidence that the generated problems are meaningful, we find that performance does not plateau: we observe monotonic improvement across all five iterations (§6.2) and continued gains when scaling to 32k questions per iteration (§6.1). If the proposer were producing meaningless or trivial specifications, we might instead expect performance to saturate.

**Diverse.** We found that functions generated as part of PSV are quite diverse yet do not recover the test questions directly. 94.3% of the generated function names in the final iteration of PSV were not found in any test benchmark. That said, only 45.3% of them were unique. Qualitatively, we found that constructed problems touch on similar themes to test questions including sorting, string operations, array quantification, and sequence reversal. We found that this diversity is important to performance; ablating the diversity mechanism we use (dynamically updating few shot sample banks) causes a 10.6% average drop in performance (Table 2).

**Non-trivial.** The solver's solve rate starts very low: 12.1% at iteration 0. These problems are certainly non-trivial. But by iteration 3, the solve rate on generated problems is up to 46.5%, showing that generated specs are challenging enough that solving them requires genuine model improvement over multiple self-play rounds, but solvable after self-play training. This difficulty-awareness is important to performance (Table 2).

## I. Generalization to Larger Models

Our main experiments use a 3B model due to limited compute. To test whether the verified solutions produced during self-play are useful beyond the 3B scale, we fine-tune three larger models (up to 32B, the largest we could fit on a single A100 80GB GPU) with LoRA SFT on the verified solutions generated by our 3B PSV-VERUS run. Results are in Table 6.

The training signal produced by PSV-VERUS transfers across model scales, benchmarks, and model families. Verified solutions from a 3B model produce large improvements on models up to $10\times$ larger (e.g., Qwen3.5-27B improves from

*Table 6.* Generalization to larger models. We LoRA-SFT each model on verified solutions generated by our 3B PSV-VERUS run, and report pass@k (%) on MBPP and HumanEval. **Baseline** is the un-tuned model; **+PSV-MBPP** and **+PSV-HE** fine-tune on PSV-VERUS-generated verified solutions for MBPP and HumanEval, respectively.

| Model | Condition | MBPP | | HumanEval | |
|---|---|---|---|---|---|
| | | pass@1 | pass@10 | pass@1 | pass@10 |
| Qwen3.5-9B | Baseline | 1.58 | 10.04 | 4.57 | 14.33 |
| | +PSV-MBPP | 31.37 | 45.70 | 18.00 | 24.58 |
| | +PSV-HE | 15.96 | 29.01 | 20.86 | 27.06 |
| Qwen3.5-27B | Baseline | 4.52 | 20.55 | 13.29 | 22.86 |
| | +PSV-MBPP | 29.04 | 42.47 | 19.43 | 31.43 |
| | +PSV-HE | 18.49 | 32.88 | 21.86 | 34.29 |
| Qwen2.5-32B-Instruct | Baseline | 3.56 | 13.70 | 9.00 | 18.57 |
| | +PSV-MBPP | 31.64 | 45.21 | 17.29 | 25.71 |
| | +PSV-HE | 18.36 | 23.29 | 19.71 | 27.14 |

*Table 7.* Reference comparison to proprietary models on verified code generation (pass@1, %). Proprietary models cannot be fine-tuned and their sizes are undisclosed, so they are reference points rather than directly comparable baselines for our training method. PSV-VERUS-32k denotes PSV-VERUS trained with 32k questions per iteration.

| Model | Params | Training | MBPP | HumanEval |
|---|---|---|---|---|
| GPT-5.2 | undisclosed | proprietary | 21.6 | 26.0 |
| Claude Opus 4.6 | undisclosed | proprietary | 60.8 | 47.9 |
| PSV-VERUS (ours) | 3B | self-play only | 36.8 | 19.1 |
| PSV-VERUS-32k (ours) | 3B | self-play only | 44.3 | 26.3 |

4.52% to 29.04% MBPP pass@1, and Qwen2.5-32B-Instruct from 3.56% to 31.64%). Training on PSV-VERUS-generated MBPP data also improves HumanEval performance (and vice versa) at every scale, indicating that PSV-VERUS teaches transferable verification skills rather than benchmark-specific solutions. The gains span both the Qwen2.5 and the newer Qwen3.5 families (the latter of which includes linear-attention layers), suggesting the training signal is not specific to a single architecture.

## J. Reference Comparison to Proprietary Models

PSV-VERUS is a training-method contribution: its central claim is that formal verification provides a sound training signal for self-play, and the most direct way to evaluate it is to measure how much it improves a model relative to other training methods on the same base model (§5). Proprietary frontier models cannot be fine-tuned, and their sizes and training data are undisclosed, so they are not directly comparable baselines for our method. Nonetheless, they provide useful reference points for contextualizing absolute performance, which we report in Table 7.

Despite being orders of magnitude smaller and trained entirely through self-play with no human-written solutions beyond the initial seed data, PSV-VERUS is competitive with frontier proprietary models on verified code generation. Our 3B model surpasses GPT-5.2 on MBPP (36.8% vs 21.6%), and with scaled-up training (32k questions per iteration) PSV-VERUS-32k reaches 44.3% on MBPP and 26.3% on HumanEval, exceeding GPT-5.2 on both benchmarks. Claude Opus 4.6 achieves the highest overall scores, but it is also a vastly larger model with access to extensive proprietary training data.

## K. Training the Proposer

In PSV-VERUS, the proposer is implemented through in-context prompting rather than as a learned model that is updated during self-play. A natural alternative is to train the proposer explicitly so that it can be optimized toward generating useful problems. We experimented with this variant, updating the proposer during self-play alongside the solver.

We found that training the proposer *reduced* the diversity of its proposals significantly and hurt downstream performance: the trained proposer collapsed toward a narrower set of problem types rather than maintaining the breadth needed for effective curriculum generation. This is consistent with our finding that diversity is crucial to performance (Table 2, −Diversity). We therefore retain the in-context proposer in PSV-VERUS. Developing a way to train the proposer while preserving proposal

diversity is an interesting direction for future work.

