# OpenReview forum: "Propose, Solve, Verify: Self-Play Through Formal Verification"
_ICML.cc/2026/Conference — ICML 2026 regular_

### Official Review · Reviewer_4cgC · 2026-02-26

**Soundness:** 3
**Presentation:** 3
**Significance:** 2
**Originality:** 2
**Overall Recommendation:** 4
**Confidence:** 5

**Summary:**

This paper proposes PROPOSE, SOLVE, VERIFY (PSV), a self-play framework for improving LLMs in verified code generation. The key idea is to leverage formal verification as a reliable correctness signal, allowing models to autonomously generate new problem specifications, solve them, and use verifier feedback to improve iteratively.

The authors apply PSV to Verus-based Rust verification, producing PSV-VERUS. Experimental results show significant improvements over AlphaVerus and iterative RFT baselines across multiple verified code benchmarks such as Dafny2Verus, MBPP, and HumanEval, achieving up to 65.63% pass@1 vs 24.06% for AlphaVerus.

**Compliance With Llm Reviewing Policy:**

Affirmed.

**Final Justification:**

We thank the authors for the thorough rebuttal. The clarification on PSV's adaptive proposer mechanism and supporting ablations adequately address our novelty concern. We acknowledge this is a meaningful distinction from prior work including SAFE.

We still encourage the authors to include a controlled SAFE baseline comparison (same base model) in the final version to strengthen the evaluation. We raise our score from 3 to 4.

**Key Questions For Authors:**

### Q1. Scaling to larger models

Have you evaluated PSV with larger (or newer) LLMs? or how does PSV compare against strong models such as GPT or Claude models?

### Q2. Quality of proposed specifications

How do you ensure generated specifications are meaningful and non-trivial? Is there risk of distribution collapse (on algorithmic tasks) or generating overly simple tasks?

### Q3. Applicability to system-level proof generation (e.g., VeruSAGE) [Additional]

The paper evaluates PSV primarily on algorithmic programming benchmarks (e.g., Dafny2Verus, MBPP, HumanEval). However, many real-world verification tasks involve system-level proofs, which require proofs beyond only loop invariants.

Can the authors comment on whether PSV is effective in such settings? Specifically:
* Have you evaluated PSV on system-level verification tasks, such as those in VeruSAGE?
* Does the proposer still generate meaningful and useful specifications at the system level, where specifications are more complex and tightly coupled?

**Limitations:**

Yes

**Strengths And Weaknesses:**

# Strengths
* Targeting on an important question
* Strong empirical improvements


# Weaknesses
* Limited conceptual novelty to prior self-play (self-evolution) and RFT frameworks
* Unsound evaluation in terms of base models, benchmarks, and baseline approaches

## 1. Limited conceptual novelty relative to prior self-play (self-evolution) and RFT frameworks

The core pipeline is essentially a combination of existing ideas:
* The "propose, solve, verify" pipeline is quite similar with SAFE (self-evolution)
* Expert iteration / rejection fine-tuning is also proposed by existing work

The main novelty appears to be the integration of formal verification into the "propose, solve, verify" loop, which is valuable but arguably an incremental extension rather than a fundamentally new paradigm.

The PSV algorithm itself is structurally very similar to existing self-evolution frameworks.

## 2. Evaluation is not sound in terms of base model, benchmark, and baseline approaches.

* The experiments use a 3B model (Qwen2.5-Coder-3B-Instruct), and it is proposed two years ago. The evaluation can be strengthened with a larger (newer) model.
* This paper compares its effectiveness with AlphaVerus (without tree refinement), without existing data synthesis baselines, e.g., SAFE (ICLR'25).
Additionally, it is better to compare PSV's effectiveness with SOTA commercial models, e.g., gpt-5.2/claude sonnet 4.5.
* It is better to add discussion about the repo/system-level proof generation capabilities, e.g., VeruSAGE.

[SAFE] Chen T, Lu S, Lu S, et al. Automated proof generation for rust code via self-evolution, ICLR 2025.

[VeruSAGE] Yang C, Neamtu N, Hawblitzel C, et al. VeruSAGE: A Study of Agent-Based Verification for Rust Systems[J]. arXiv preprint arXiv:2512.18436, 2025.

---

> ### Author Rebuttal · Authors · 2026-03-30
>
> Thank you for your review and for recognizing that our work targets an important question and achieves strong empirical improvements. We address your concerns and questions below.
>
> Before we do, we’d like to comment that this was a particularly thoughtful and insightful review. We’re grateful, and we hope we address your concerns satisfactorily below.
>
> > The "propose, solve, verify" pipeline is quite similar with SAFE
>
> At a high level, PSV and SAFE investigate different questions. PSV investigates whether a proposer can propose *new* problems in an *adaptive* way, while SAFE generates specifications based on existing Rust programs without an adaptive mechanism. More specifically:
>
> 1. SAFE **doesn't propose new problems**, but operates on a **fixed set of existing Rust programs**. PSV's proposer generates entirely new formal specifications, requiring several novelties: (a) ensuring diversity via iteratively updating the prompt (Sec 3.1), (b) controlling difficulty via pass-rate based labels (Sec 3.1), (c) filtering invalid specifications through compilation (Appendix D).
> 2. SAFE trains for proof annotation (generating a correct proof given correct existing code). PSV trains for the more general task of end-to-end formally verified code generation (both code and proof together).
> 3. In PSV, the proposer and solver co-evolve: the proposer adapts problem difficulty based on solver's pass rate each iteration. In SAFE, specification synthesis and proof synthesis are decoupled.
>
> We will include this discussion in the revised paper.
>
> > Expert iteration / rejection fine-tuning is also proposed by existing work
>
> We agree, and do not claim RFT as our contribution. RFT is the training mechanism we use, and could be replaced with other learning algorithms (e.g., on-policy RL). Our contribution is the self-play framework that generates and expands a difficulty-based training curriculum in the formally verified code setting.
>
> > The main novelty appears to be the integration of formal verification ... which is valuable but arguably an incremental extension
>
> We respectfully disagree. We introduced several new components to ensure the stability of this iterative process including difficulty-aware and diversity-aware proposal mechanisms and specification compilation for filtering (each validated through ablations in Table 2). Further, we show empirically that formal verification overcomes a key obstacle in self-play for code: "imperfect verifiers such as incomplete unit tests can be exploited by the solver, limiting improvement".
>
> > The experiments use a 3B model … the evaluation can be strengthened with a larger (newer) model.
>
> We used a 3B model due to limited compute. We have included experiments with newer, larger models in our response to reviewer AWnC above.
>
> > This paper compares with AlphaVerus (without tree refinement), without existing data synthesis baselines, e.g., SAFE (ICLR'25). Additionally, it is better to compare PSV's effectiveness with SOTA commercial models
>
> Our contribution is **a method for improving a verified code model through self-play, not the model itself**. The appropriate comparison is against other training methods applied to the same base model (AlphaVerus, RFT), which we outperform significantly. We have added SOTA commercial model comparisons in our response to reviewer AWnC above, though these are somewhat orthogonal to the purpose of our experiments.
>
> Regarding SAFE: it addresses a different task (proof generation for existing code, not proposer-solver self-play), uses DeepSeekCoder-33B (11× larger), and relies on GPT-4o for bootstrapping, whereas PSV requires no proprietary model. A direct comparison would conflate model capacity with training methodology.
>
> > It is better to add discussion about the repo/system-level proof generation capabilities, e.g., VeruSAGE.
>
> VeruSAGE is concurrent work first made public one month before the ICML submission, so we consider it out of scope given ICML's concurrent boundary policy. We view algorithmic problems as the appropriate starting point for studying self-play in this domain, in line with prior work (AlphaVerus, SAFE).
>
> That said, extending PSV to system-level verification is an exciting future direction. PSV's core principles — formal verification as a sound training signal, difficulty-aware proposal, and iterative self-play — are not specific to function-level tasks. The self-play loop could combine with agentic scaffolding (as in VeruSAGE) for multi-step proof strategies. We will add this discussion in the revised paper, and thank you for an insightful comment.
>
> > How do you ensure generated specifications are meaningful and non-trivial? Is there risk of distribution collapse (on algorithmic tasks) or generating overly simple tasks?
>
> We've addressed this in our response to reviewer FqBt above, and direct the reviewer there due to space constraints.

---

> > ### Author Rebuttal · Reviewer_4cgC · 2026-04-03
> >
> > We thank the authors for the thorough rebuttal. The clarification on PSV's adaptive proposer mechanism and supporting ablations adequately address our novelty concern. We acknowledge this is a meaningful distinction from prior work including SAFE.
> >
> > We still encourage the authors to include a controlled SAFE baseline comparison (same base model) in the final version to strengthen the evaluation. We raise our score from 3 to 4.

---

### Official Review · Reviewer_AWnC · 2026-03-12

**Soundness:** 2
**Presentation:** 2
**Significance:** 2
**Originality:** 2
**Overall Recommendation:** 3
**Confidence:** 3

**Summary:**

This paper presents a self-play framework  PSV  for training large language models to generate code for formal verification without human supervision. It leverages a formal verifier to provide reliable correctness signals, driving an iterative cycle of "proposer-solver".  The proposer generates new specifications, the solver attempt to generate code and proofs that meet the specifications and  the verifier verify the solutions to ensure mathematical guarantees of correctness. The authors traine PSV-VERUS within this framework, achieving improvements over AlphaVerus and RFT baselines on the Dafny2Verus, MBPP, and HumanEval benchmarks

**Compliance With Llm Reviewing Policy:**

Affirmed.

**Final Justification:**

Please see my comments.

**Key Questions For Authors:**

In my weakness.

**Strengths And Weaknesses:**

## Strength

**S1.** The authors propose to adopt  formal verification for code generation. The reliable verification signals provide correctness guarantees in a mathematics.

**S2.** The authors claim to open-source code and model weights.


## Weakness

**W1.** All the experiments are conducted based on the Qwen2.5-Coder-3B-Instruct, a small language model. Therefore, its training performance are significantly different from those of large-scale models such as 13B, 70B, or MOE models. The contribution of this paper is limited.

**W2.** The author should add more baselines, some closed-source strong model, like Opus-4.6, GPT-5.3 ....

---

> ### Author Rebuttal · Authors · 2026-03-30
>
> We thank the reviewer for their feedback. We address each concern below.
>
> ### W1: Experiments on different models
> We agree that evaluating on larger models strengthens the paper. We fine-tuned four larger models (up to 32B, the largest we could fit on a A100 80GB GPU). We trained with LoRA SFT using verified solutions generated by our 3B model during PSV training to see whether the verified code produced during PSV improves performance for larger models as well.
>
> | Model | Condition | MBPP p@1 | MBPP p@10 | HE p@1 | HE p@10 |
> |-------|-----------|----------|-----------|--------|---------|
> | Qwen3.5-9B | Baseline | 1.58% | 10.04% | 4.57% | 14.33% |
> | Qwen3.5-9B | +PSV-MBPP | **31.37%** | **45.70%** | 18.00% | 24.58% |
> | Qwen3.5-9B | +PSV-HE | 15.96% | 29.01% | **20.86%** | **27.06%** |
> | Qwen3.5-27B | Baseline | 4.52% | 20.55% | 13.29% | 22.86% |
> | Qwen3.5-27B | +PSV-MBPP | **29.04%** | **42.47%** | 19.43% | 31.43% |
> | Qwen3.5-27B | +PSV-HE | 18.49% | 32.88% | **21.86%** | **34.29%** |
> | Qwen2.5-32B-Instruct | Baseline | 3.56% | 13.70% | 9.00% | 18.57% |
> | Qwen2.5-32B-Instruct | +PSV-MBPP | **31.64%** | **45.21%** | 17.29% | 25.71% |
> | Qwen2.5-32B-Instruct | +PSV-HE | 18.36% | 23.29% | **19.71%** | **27.14%** |
>
> These results show that PSV's training signal **transfers across model scales, benchmarks, and model families**. Verified solutions from a 3B model produce large improvements on models up to 10x larger (e.g., Qwen3.5-27B: 4.52% → 29.04% MBPP p@1; Qwen2.5-32B-Instruct: 3.56% → 31.64%). Training on PSV-MBPP data also improves HumanEval performance (and vice versa) at every scale, indicating PSV teaches transferable verification skills rather than benchmark-specific solutions. The gains span both Qwen2.5 and the newer Qwen3.5 families, confirming the signal is not architecture-specific (Qwen3.5 have linear attention layers as well).
>
> ### W2: Closed-Source Model Baselines
>
> Thank you for this suggestion. We note that PSV is a **training algorithm** contribution — the central claim is that formal verification provides a superior training signal for self-play. The most direct way to evaluate a training algorithm is to measure how much it improves a model's performance, which is what our experiments demonstrate. Many closed-source models cannot be fine-tuned and the model sizes are unknown, so they are not fair baselines for comparison with our method.
>
> That said, we agree that closed-source models provide useful reference points for contextualizing absolute performance. We have run these experiments and include the results of pass@1 below.
>
> | Model | Params | Training | MBPP | HumanEval |
> |-------|--------|----------|------|-----------|
> | GPT-5.2 | undisclosed | proprietary | 21.6% | 26.0% |
> | Claude Opus 4.6 | undisclosed | proprietary | 60.8% | 47.9% |
> | PSV-Verus (ours) | 3B | self-play only | 36.8% | 19.1% |
> | PSV-Verus-32k (ours) | 3B | self-play only | 44.3% | 26.3% |
>
> Despite (likely) being orders of magnitude smaller and trained entirely through self-play with no human-written solutions beyond the initial seed data, PSV-Verus achieves performance competitive with frontier closed-source models on verified code generation. Our 3B-parameter model surpasses GPT-5.2 on MBPP (36.8% vs 21.6%). With scaled-up training (32k questions per
> iteration), PSV-Verus-32k reaches 44.3% on MBPP and 26.3% on HumanEval — exceeding GPT-5.2 on both benchmarks. While Claude Opus 4.6 achieves the highest overall scores, it is a vastly larger model with access to extensive proprietary training data.
>
> ### W3: Benchmark Difficulty and SWEbench
> We would first like to clarify our problem setting, and explain why we chose these benchmarks. We are interested in studying self-play in verified code generation, because in verified code generation, proposed problems *are defined by* their verification criteria (e.g., requires, ensures…etc), making them theoretically immune to reward hacking – a critical bottleneck in self play methods in the past. Within formally verified code generation, MBPP-Verified and HumanEval-Verified are standard benchmarks, used by both AlphaVerus and SAFE (ICLR'25). SWE-bench and terminal-bench are not formally verified, and so are orthogonal to this goal and out of scope.
>
> We would also respectfully like to push back on the benchmarks being “too easy”. **The benchmarks are not easy.** The prior SOTA (AlphaVerus) achieves only 24% / 6.5% / 7.2% pass@1 on Dafny2Verus / MBPP-Verified / HumanEval-Verified respectively, and even after PSV training no benchmark is saturated. While MBPP in Python achieves 50–80% pass@1 for similar-scale models; the MBPP-**Verified** variant is much lower because formal verification transforms standard tasks into challenging problems requiring loop invariants, ghost annotations, and mathematical proofs of correctness.

---

> > ### Author Rebuttal · Reviewer_AWnC · 2026-04-01
> >
> > Thanks for your response, I have changed my score.

---

> > > ### Author Response · Authors · 2026-04-04
> > >
> > > Thank you for your response. We valued your feedback, and your suggestions are making the paper better with the extended experiments. Your final rating for the paper can have a significant impact on the final decision. Could you please let us know if there are any other concerns that are needed for you to raise our score, since you indicated that we have addressed all of your existing concerns. Understanding any additional concerns you might have would help us strengthen our work.

---

### Official Review · Reviewer_iNJv · 2026-03-12

**Soundness:** 2
**Presentation:** 3
**Significance:** 3
**Originality:** 3
**Overall Recommendation:** 4
**Confidence:** 4

**Summary:**

The authors present a framework that enables LLMs to improve through self-play: The model is prompted to propose challenging coding problems to verify and then to verify them. The feedback from successes in this framework is leveraged for RL. They demonstrate remarkable performance on a set of three verification problems.

**Compliance With Llm Reviewing Policy:**

Affirmed.

**Final Justification:**

Two of my main concerns remain unaddressed. The paper is interesting but there is a real chance that the results don't transfer so I cannot whole-heartedly recommend acceptance.

**Key Questions For Authors:**

- What is the improvement over the base model?
- Was AlphaVerus retrained on the base model or did you use the model trained by the AlphaVerus authors?
- Are there any similarities between the self-proposed methods in the Transfer Learning Setup and the HumanEval/MBPP functions?
- Figure 4 and 5 show no signs of saturation for the range of parameters presented. How long does this trend go on? It would be interesting to see if this plateaus at some point (possibly limited by model capabilties).
- Do your findings transfer to other LLMs?

**Limitations:**

There is no discussion of limitations, but I think there are some:
- Unclear how general the findings are across LLMs (sizes, architectures)
- A human-written set of initial problems is still required, no complete bootstrapping (including the pre-training)
- Scaling trends: Are there fundamental limitations to scale this method up or does it hit limits at some point?

**Strengths And Weaknesses:**

Strengths:

- *Significance*: The authors tackle a highly relevant problem (improving LLM capabilities). Their experiments show that improvement in the formal verification domain is possible and their ablations shine interesting light on the nuances needed to achieve them (for example that the categorization into easy/hard/etc is necessary and diversifying the prompt has a huge impact)
- *Presentation*: The presentation is overall clean and the text is well written and precise.

Weaknesses:
- *Soundness*: The performance of the base model before training should be presented in Table 1, so that the overall impact of the training can be assessed more clearly.
- *Soundness*: It is unclear whether AlphaVerus was retrained or the authors used the model produced by the AlphaVerus authors. In the latter case the outputs are hard to comapare because AlphaVerus uses a different base model.
- *Soundness*: Even though HumanEval and MBPP are considered held-out test cases, the model might propose them during training because the functions are well known and possibly leaked into the training data of the LLM.
- *Significance*: The authors present their results only on a single LLM. It would be interesting to see if they generalize.

Typo in line 412: The difference

---

> ### Author Rebuttal · Authors · 2026-03-30
>
> We thank Reviewer iNJv for the thoughtful and constructive review. We are glad the reviewer recognizes the significance of the problem, the quality of the ablations, and the clean presentation. Below, we address each concern in detail.
>
> ---
>
> ### Weakness 1/ Question 1: Base Model Performance Should Be in Table 1
> We agree that showing the base model performance makes the training impact clearer. We’ve listed the performance below, and have added it to Table 1.
>
> Our base model results are below (and will be added to the paper):
> | | Dafny2Verus | | | MBPP | | | HumanEval | | |
>   |---|---|---|---|---|---|---|---|---|---|
>   | | pass@1 | pass@5 | pass@10 | pass@1 | pass@5 | pass@10 | pass@1 | pass@5 | pass@10 |
>   | **Base Model** | 8.05₀.₁₉ | 20.68₀.₅₈ | 28.15₁.₀₃ | 0.87₀.₁₂ | 4.02₀.₄₅ | 7.53₀.₇₇ | 4.55₀.₂₃ | 10.28₀.₄₂ | 12.86₀.₃₇ |
>
>   The base model baseline is substantially weaker across all metrics than the AlphaVerus baseline we listed: 8.05 vs 24.06 on Dafny2Verus pass@1, 0.87 vs 6.48 on MBPP pass@1, and 4.55 vs 7.24 on HumanEval pass@1.
>
>
> ---
>
> ### Weakness 2 / Question 2: Was AlphaVerus Retrained or Did You Use the Original Authors' Model?
>
> **We re-implemented the AlphaVerus methodology on the same base model (Qwen2.5-Coder-3B-Instruct) used for PSV-Verus**, rather than using the original AlphaVerus authors' model (which uses a different base model). This ensures the comparison isolates the effect of the training methodology. Specifically, our AlphaVerus baseline reuses publicly available code (the “without treefinement version”): 50 in-context examples with no fine-tuning, applied to our base model. We acknowledge this was not stated explicitly in the paper, and have added it.
>
> ---
>
> ### Weakness 3/ Question 3: Data Contamination — Could HumanEval/MBPP Be Proposed During Training?
> This is an interesting question. We ran an analysis on the generated problems and found that there were no exact matches between generated and heldout test specs, across any seeds of our experiment, across any of our experiments (even including the scaling experiments, to 32k questions). That said, it is not impossible that some of the problems ended up being similar.
>
> ---
>
> ### Q4: Do Figures 4 and 5 Show Saturation? How Long Does the Trend Continue?
> We appreciate that you noticed this! This is a great question, and one we've thought about a lot. Unfortunately this is outside our compute budget to test further, as the x axis is log scale. We are interested in expanding out these graphs in future work. The short answer is: the graphs don't seem to show saturation, and we don't know how high the upside is or could be. We think analyzing scaling laws over a method like this would be very interesting.
>
> ### W4/Q5: Do Findings Transfer to Other LLMs?
> We’ve run an additional experiment on this, in our response to reviewer AWnC.
>
> ### W5: Limitations Discussion
> Yes this is great feedback, thank you. We'll add a discussion with each of these points. We appreciate the detailed review of our work!
>
> ### W6: Typo in Line 412
> Thank you for catching this. We have corrected the typo in the revised manuscript.

---

> > ### Author Rebuttal · Reviewer_iNJv · 2026-04-03
> >
> > I thank the authors for their extensive answers. I think two main concerns remain:
> > - There is no verification of the generality of this training recipe on models outside the Qwen family (for example on Olmo). I can forgive this as it seems a general trend in training LLMs with RL, but I don't like this trend.
> > - The authors checked contamination using exact match. I think a more robust check using embeddings or close matches is warranted since LLMs are known to be very capable of transferring to highly similar settings.

---

### Official Review · Reviewer_FqBt · 2026-03-14

**Soundness:** 2
**Presentation:** 3
**Significance:** 2
**Originality:** 2
**Overall Recommendation:** 4
**Confidence:** 4

**Summary:**

This paper proposes a self-play framework for improving language models on formally verified code generation. In this framework, a proposer generates new formal specifications, a solver attempts to produce implementations that satisfy them, and a formal verifier checks correctness. Verified solutions are then used to fine-tune the solver, while solver performance is used to guide the generation of new problems. The resulting model PSV-VERUS is verified on Rust programming and improves performance on several verified code generation benchmarks compared with prior baselines.

**Compliance With Llm Reviewing Policy:**

Affirmed.

**Key Questions For Authors:**

Since the specifications are generated by an LLM proposer, can the authors provide more analysis on the quality of these specifications (e.g., diversity, difficulty, or semantic meaningfulness)? This would help clarify whether the proposed framework truly expands the problem distribution rather than generating minor variations.

The proposer is implemented through in-context prompting rather than a learned model. Have the authors considered training the proposer explicitly or optimizing it during the self-play process?

The improvements appear stronger in the test-time training setting than in the transfer setting. Can the authors provide additional analysis or discussion on why the transfer improvements are relatively limited, and whether the method can improve broader generalization?

**Limitations:**

Yes.

**Strengths And Weaknesses:**

# Strengths
1. The paper studies self-improving LLMs through self-play for verified code generation, which is an interesting problem and an important direction.
2. The paper is well written and easy to follow, and the overall framework is clearly described.
3. The technical approach is reasonable and the methods used are generally sound.

# Weaknesses
1. The paper does not analyze the semantic quality of the LLM-generated specifications. While the verifier ensures correctness with respect to the specification, the paper does not examine whether the generated specifications are meaningful, diverse, or non-trivial, making it unclear whether the proposed curriculum generation truly expands the problem space or mainly produces minor variations of existing specifications.
2. Improvements are substantially stronger in the training setting than in the transfer setting, suggesting limited evidence of broader generalization. While the method achieves large gains on datasets involved in the self-play training process, the improvements on transferred benchmarks are more modest, raising the question of whether the approach mainly adapts to the distribution of generated specifications rather than improving more general reasoning capabilities.

---

> ### Author Rebuttal · Authors · 2026-03-30
>
> We thank the reviewer for recognizing the importance of self-improving LLMs through self-play for verified code generation, and for the positive feedback on our writing. We address each concern below.
>
> ### Semantic Quality of Generated Specifications
> > the paper does not examine whether the generated specifications are meaningful, diverse, or non-trivial
>
> This is a good point, and very similar to one raised by Reviewer 4cgC. We analyze generated specifications to see whether they are **meaningful**, **diverse**, and **non-trivial** — with quantitative evidence summarized below. We will update the paper with this discussion.
>
> **Meaningful.** In thinking about whether the generated specs are meaningful, we should clarify that all generated specifications pass through a multi-stage filtering pipeline that enforces syntactic validity through spec verification. 72.5% of raw generations fail spec verification (compilation + type-checking against the Verus compiler), and after further deduplication, only 5.4% of the 10k raw proposals per iteration survive as unique valid specifications (~540 per iteration). Every surviving spec therefore defines well-formed `ensures`/`requires` clauses over valid Verus types. Additionally, if specs were not meaningful, we would expect that performance of the method would likely plateau — instead we observe monotonic improvement across 5 iterations (Figure 4) and consistent scaling to 32k questions (Figure 5).
>
> **Diverse.** We analyzed the diversity of function specs. We found that functions generated as part of PSV are quite diverse yet do not recover the test questions directly. 94.3% of the generated function names in the final iteration of PSV were not found in any test benchmark. That said, only 45.3% of them were unique. Qualitatively, we found that constructed problems touch on similar themes to test questions including sorting, string operations, array quantification, and sequence reversal. We found that this diversity is important to performance; ablating the diversity mechanism we use (dynamically updating few shot sample banks) causes a 10.6% average drop in performance (Table 2). This is an area we will indicate in the paper could be productive for future study.
>
> **Non-trivial.** Our goal was to generate problems that were solvable but non trivial. The solver's solve rate starts very low: 12.1% at iteration 0. These problems are certainly non-trivial. But by iteration 3, the solve rate on generated problems is up to 46.5%, showing that generated specs are challenging enough that solving them requires genuine model improvement over multiple self-play rounds, but solvable after self-play training. This difficulty-awareness is important to performance: we found that ablating difficulty labels causes a 5.6% drop (Table 2).
>
> ### Training vs. Transfer Performance
> > Improvements are substantially stronger in the training setting than in the transfer setting
>
> The performance gap is largely explained by domain differences. HumanEval-Verified is substantially harder than MBPP-Verified, and its problems are far longer on average, partly because its functions rely far more on helper spec functions used in defining the target function spec. With larger benchmarks (these are only 73–74 problems each) and more compute (Figures 4–5 show no saturation up to 32k questions), we expect generalization would be more pronounced.
>
> ### Training the Proposer
> We experimented with training the proposer explicitly during self-play. This **reduced the distribution of proposals significantly and hurt performance** — the trained proposer collapsed toward a narrower set of problem types rather than maintaining the diversity needed for effective curriculum generation, consistent with our finding that diversity is crucial. We will add these results to the paper. An interesting direction for future work is developing a way to train a proposer.

---

### Decision · Program_Chairs · 2026-04-30

**Decision:**

Accept (regular)

**Comment:**

This paper proposes a self-play framework for verified code generation, which received three weak accepts and one weak reject for its clear presentation and significant empirical improvements over the baseline. The reviewers collectively acknowledge the importance of the problem and the soundness of using formal verification for reliable self-improvement signals. While valid concerns were raised regarding the limited model scale (3B parameters) and the lack of comparison to certain baselines (e.g., commercial models or SAFE), the substantial performance gains and the successful integration of formal verification into the self-play loop represent a meaningful contribution to the field. During the rebuttal stage, the author made reponse regarding model size, and provide more experimental results from ~30B models to demonstrate the generalization ability. I recommend a weak acceptance.